# Fluoropyrimidine combination therapy versus fluoropyrimidine monotherapy for gemcitabine-refractory advanced pancreatic cancer: A systematic review and meta-analysis of randomized controlled trials

**Wei Tian**[1], **Lina Zhang**[2], **Xiao Liu**[1], **Xiao Ma**[3], **Rui Wang**[1]*

**1** Department of Oncology III, Zibo Central Hospital, Shandong University, Zibo, Shandong, Peoples' Republic of China, **2** Department of Nursing, Zibo Central Hospital, Shandong University, Zibo, Shandong, Peoples' Republic of China, **3** Department of Internal Medicine, Zhangqiu People's Hospital, Zhangqiu, Shandong, People' Republic of China

\* wangrui2023@126.com

## Abstract

### Objectives

Fluoropyrimidine-based regimens have been investigated as the second line chemotherapy in patients with advanced pancreatic cancer refractory to gemcitabine. We conducted this systematic review and meta-analysis to evaluate the efficacy and safety profile of fluoropyrimidine combination therapy versus fluoropyrimidine monotherapy in such patients.

### Methods

The databases of MEDLINE, EMBASE, Cochrane Central Register of Controlled Trials, ASCO Abstracts and ESMO Abstracts were systematically searched. Randomized controlled trials (RCTs) that compared fluoropyrimidine combination therapy versus fluoropyrimidine monotherapy in patients with gemcitabine-refractory advanced pancreatic cancer were included. The primary outcome was overall survival (OS). Secondary outcomes included progression-free survival (PFS), overall response rate (ORR) and serious toxicities. Statistical analyses were performed by using Review Manager 5.3. Egger's test was performed to assess the statistical evidence of publication bias by using stata 12.0.

### Results

A total of 1183 patients from six randomized controlled trials were included for this analysis. Fluoropyrimidine combination therapy increased ORR [RR 2.82 (1.83–4.33), p<0.00001] and PFS [HR 0.71 (0.62–0.82), p<0.00001], without significant heterogeneity. Fluoropyrimidine combination therapy improved OS [HR 0.82 (0.71–0.94), p = 0.006], with significant heterogeneity (I² = 76%, p = 0.0009). The significant heterogeneity might have been caused by the different administration regimens and baseline characteristics. Peripheral neuropathy

**Data Availability Statement:** All relevant data are within the manuscript and its Supporting Information files.

**Funding:** The author(s) received no specific funding for this work.

**Competing interests:** The authors have declared that no competing interests exist.

and diarrhea were more common in the regimens containing oxaliplatin and irinotecan, respectively. No publication bias was detected by Egger's tests.

## Conclusions

Compared with fluoropyrimidine monotherapy, fluoropyrimidine combination therapy had a higher response rate and longer PFS in patients with gemcitabine-refractory advanced pancreatic cancer. Fluoropyrimidine combination therapy could be recommended in the second line setting. However, due to concerns about toxicities, the dose intensities of chemotherapy drugs should be carefully considered in patients with weakness.

## Introduction

Pancreatic cancer is associated with a poor prognosis and the seventh leading cause of cancer-related death worldwide, with 432,242 deaths reported in 2018 [1].

Gemcitabine-based chemotherapy has been the standard treatment for locally advanced and metastatic pancreatic cancer since 1997, based on modest improvement on survival when compared with fluorouracil [2]. More recently, FOLFIRINOX (a combination of oxaliplatin, irinotecan, fluorouracil and folinic acid) and gemcitabine plus nab-paclitaxel have showed superiority over gemcitabine alone in appropriately selected patients with advanced disease in phase III trials [3,4]. These two regimens have emerged as new first-line treatment options.

However, pancreatic cancer often presents with a substantial deterioration in quality of life and performance status. And FOLFIRINOX was accompanied by a substantial increase in toxicity. As such, it is demonstrated that many patients with advanced pancreatic cancer will not be eligible for this regimen and will likely receive first-line gemcitabine-based therapy.

Progression after first line therapy is inevitable in advanced pancreatic cancer, leaving clinicians with few options. In order to improve the prognosis of such patients, there is an urgent need to establish an effective second line therapy with low toxicity and high efficacy. Over the past several years, fluoropyrimidine-based regimens have been investigated as the second line chemotherapy in patients who failed in gemcitabine alone or gemcitabine-based treatment, but with diverse results. Therefore, we have undertaken this systematic review and meta-analysis to evaluate the available evidence from the relevant randomized controlled trials (RCTs).

## Materials and methods

### Search strategy

We searched MEDLINE, EMBASE, Cochrane Central Register of Controlled Trials, ASCO Abstracts and ESMO Abstracts up to January 2022 to identify the eligible studies. Our search strategy was detailed in Table 1. We used Cochrane highly sensitive search strategy to identify randomized controlled trials in MEDLINE (Ovid format). And the strategy was adapted in other databases.

All the randomized controlled trials on drug therapy or chemotherapy for patients with advanced pancreatic cancer who had experienced progression during first-line gemcitabine-based chemotherapy were collected and identified.

### Inclusion criteria

We collected all phase II–III and prospective randomized controlled trials. The criteria for inclusion were as follows: (1) Type of participants: adults with locally advanced or metastatic

**Table 1. Search strategy for MEDLINE (Ovid format) used in this meta-analysis.**

| | |
|---|---|
| 1. randomized controlled trial.pt. | 19. (pancreas adj5 neoplasm$).mp. |
| 2. controlled clinical trial.pt. | 20. (pancreatic adj5 carcinoma$).mp. |
| 3. randomized.ab. | 21. (pancreas adj5 tumor$).mp. |
| 4. placebo.ab. | 22. (pancreas adj5 tumour$).mp. |
| 5. drug therapy.fs. | 23. or/12-22 |
| 6. randomly.ab. | 24. exp drug therapy/ |
| 7. trial.ab. | 25. chemothera$.tw. |
| 8. groups.ab. | 26. drug therap$.tw. |
| 9. or/1-8 | 27. antineoplastic$.tw. |
| 10. humans.sh. | 28. or/24-27 |
| 11. 9 and 10 | 29. gemcitabine |
| 12. exp pancreatic neoplasms/ | 30. GEM |
| 13. (pancreatic adj5 cancer$).mp. | 31. Gemzar |
| 14. (pancreatic adj5 neoplasm$).mp. | 32. or/29-31 |
| 15. (pancreatic adj5 carcinoma$).mp. | 33. 23 and 28 |
| 16. (pancreatic adj5 tumor$).mp. | 34. 32 and 33 |
| 17. (pancreatic adj5 tumour$).mp. | 35. 11 and 34 |
| 18. (pancreas adj5 cancer$).mp. | |

Demonstration for characters used in the search strategy. pt: Publication type. ab: Abstract. fs: Floating subheading. sh: Subheading. adj: Adjacent. mp: Indicates a search of title, original title, abstract, name of substance word and subheading word. tw: Text word. $: Truncation operator.

pancreatic cancer who developed progression after previous gemcitabine-based therapy in a neoadjuvant or adjuvant setting. (2) Type of study: studies had to be RCTs comparing fluoropyrimidine combination therapy versus fluoropyrimidine monotherapy in patients with gemcitabine-refractory pancreatic cancer. Gemcitabine-refractory was defined as follows: (1) Participants confirmed advanced or metastatic pancreatic cancer who had experienced progression during or after the first-line chemotherapy including gemcitabine. (2) Participants with pancreatic cancer who relapsed within 6 months after completing gemcitabine-based chemotherapy as adjuvant or neoadjuvant.

## Exclusion criteria

We excluded quasi-randomized studies which possessed insufficient quality. Cross-over studies were also excluded in order to evaluate the overall effect on OS. Leucovorin or folinic acid was considered as chemotherapy sensitizer for fluoropyrimidine. So fluoropyrimidine plus leucovorin or folinic acid was seen as fluoropyrimidine monotherapy. Studies of fluoropyrimidine plus leucovorin or folinic acid compared with fluoropyrimidine were also excluded. Two reviewers (Wei Tian and Rui Wang) independently screened each record and each report retrieved. And two other reviewers (Lina Zhang and Xiao Liu) independently analyzed all the details of each article to confirm that they met the inclusion criteria.

## Data extraction

Two reviewers (Wei Tian and Rui Wang) independently extracted the data from all eligible studies. When discrepancies arose, a third reviewer was needed to make the final decision. Conflicts were resolved through discussion. The primary outcome was overall survival (OS). Secondary outcomes included progression-free survival (PFS), overall response rate (ORR)

and serious toxicities. We analyzed hazard ratio (HR) of time-to-event data for OS and PFS, as provided by Jayne F Tierney et al [5]. In addition, we analyzed risk ratio (RR) of dichotomous data for ORR and toxicities. The HRs were extracted from the original studies or accounted from the reported events and the corresponding p-value of the log-rank statistics, or by reading off survival curves. We used Jadad score to assess the methodological quality of the included studies. Three items (description of randomization, double blindness and withdrawals) were directly related to the quality of studies. When randomization, double blindness and withdrawals were mentioned in the study, 1 score for each item was given. When appropriate methods of randomization or double blindness was described, another 1 score for each item was given. Each eligible study was evaluated and given a score from 0 to 5. Missing data would be obtained through contacting the authors or reading off the Kaplan-Meier curves.

## Statistical analysis

Review Manager 5.3 software was used to perform the statistical analysis on the data. Time-to-event outcomes were compared using HR. Dichotomous data were compared using risk ratio (RR). 95% confidence intervals (CI) were calculated for each estimate. Statistical heterogeneity was assessed by the chi-square test, and expressed by the $I^2$ index. The fixed-effect model weighted by the Mantel-Haenszel method was used. Further analysis (subgroup analysis, sensitivity analysis or random-effect model) was performed to identify the potential cause when considerable heterogeneity was found ($p < 0.1$, or $I^2 > 50\%$). Furthermore, we used Stata 12.0 to perform Egger's test in order to assess the publication bias.

## Results

### Study identification

Our search screened 95 trials, and found 6 publications related to 6 randomized controlled trials (1183 patients) that compared fluoropyrimidine combination therapy versus fluoropyrimidine monotherapy in patients with gemcitabine-refractory pancreatic cancer. Two phase II [6,7] and four phase III [8–11] trials were included. 3 of the trials compared fluoropyrimidine and oxaliplatin with fluoropyrimidine monotherapy [6,8,10]. 2 of the trials compared fluoropyrimidine and irinotecan with fluoropyrimidine monotherapy [7,9]. 1 of the trials compared fluoropyrimidine, oxaliplatin and irinotecan with fluoropyrimidine monotherapy [11]. A diagram representing the flow of the identification of the RCTs was shown in **Fig 1**.

### Characteristics of included studies

Characteristics of patients included in the studies were shown in **Table 2**. The distribution of baseline for patient characteristics was found to be a little inconsistent, mainly caused by Gill 2016 [8]. We found that Wang 2015 had 117 and 149 patients in the 2 arms, which indicated some imbalance in the randomization [9]. The study of Wang 2015 had 3 arms. The arm consisting of nanoliposomal irinotecan and fluorouracil was added after the protocol amendment. 30 patients were assigned to fluoropyrimidine monotherapy arm before all sites switched to the new protocol. Regimens and endpoints of included studies were shown in **Table 3.** Methodological details which might cause bias were shown in **Table 4**. All the 6 studies included in this meta-analysis were open-label, and included illustrations regarding randomization. 4 of the trials described the detailed methods used for randomization [6,9–11]. All the 6 trials reported detail information of withdrawals. PFS was defined as the time from randomization to the disease progression or death from any cause. OS was defined as the time from randomization to death from any cause. There were no differences in the definitions of PFS and OS

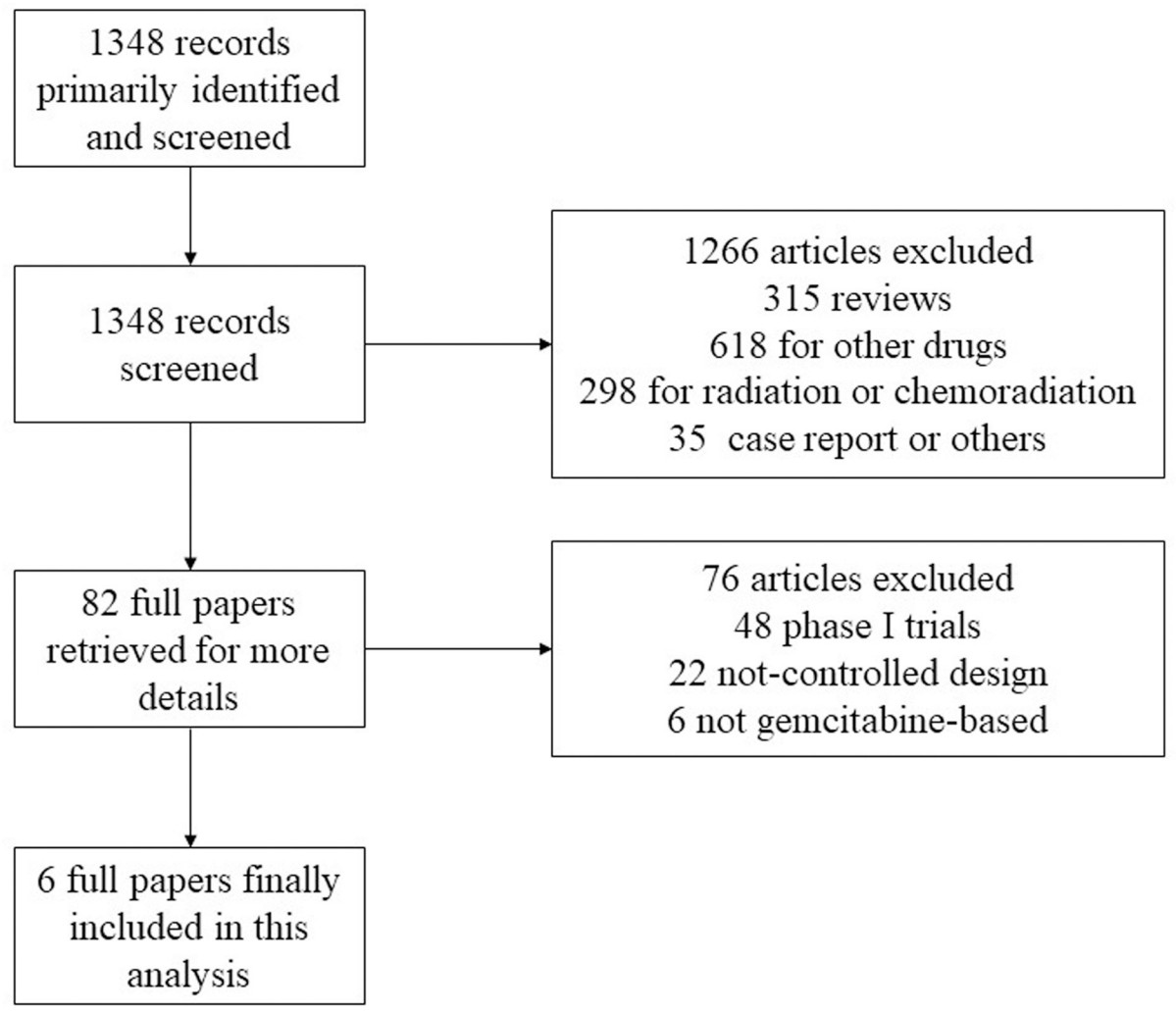

**Fig 1. Flow chart for identification and inclusion of trials for this meta-analysis.**

**Table 2. Patients characteristics of included studies.**

| References | Description for GEM-refractory | Patients enrolled | Male (%) | ECOG ≤1 or KPS ≥70 (%) | Median age (years) |
|---|---|---|---|---|---|
| Ohkawa et al. 2015 [6] | progression during first-line GEM or relapsed within 24 weeks after adjuvant GEM | 136 / 135 | 61.2 / 61.5 | 100 / 100 | 65 / 63.5 |
| Ioka et al. 2017 [7] | progression during first-line GEM or relapsed within 24 weeks after adjuvant GEM | 67 / 70 | 58.3 / 67.2 | 100 / 100 | 62 / 65 |
| Gill et al. 2016 [8] | progression during or after first-line GEM | 54 / 54 | 57.4 / 55.6 | 88.9 / 94.4 | 65 / 67 |
| Wang et al. 2015 [9] | progression during or after first-line GEM or relapsed within 6 months after adjuvant/neoadjuvant GEM | 117 / 149 | 59 / 56 | 97 / 99 | 63 / 62 |
| Oettle et al. 2014 [10] | progression during first-line GEM | 77 / 91 | 52.6 / 57.1 | 100 / 100 | 62 / 61 |
| Se-Il et al. 2021 [11] | progression during or after first-line GEM or relapsed within 6 months after adjuvant GEM | 41 / 41 | 72 / 68 | 100 / 100 | 59 / 62 |

GEM: Gemcitabine. ECOG: Eastern Cooperative Oncology Group. KPS: Karnofsky Performance Status.

**Table 3. Regimens and endpoints of included studies.**

| References | Regimens (per arm) | Interventions | Primary endpoint |
|---|---|---|---|
| Ohkawa et al. 2015 [6] | S-1+OXA<br>S-1 | Arm A: OXA 100 mg/m$^2$ iv d1, S-1 80/100/120 mg/d based on BSA, po d1–14, q3w.<br>Arm B: S-1 80/100/120 mg/d based on BSA, po d1–28, q6w. | PFS |
| Ioka et al. 2017 [7] | S-1+IRI<br>S-1 | Arm A: IRI 100 mg/m$^2$ iv d1 d15, S-1 80/100/120 mg/d based on BSA, po d1–14, q4w.<br>Arm B: S-1 80/100/120 mg/d based on BSA, po d1–28, q6w. | PFS |
| Gill et al. 2016 [8] | FU/LV+OXA<br>FU/LV | Arm A: OXA 85 mg/m$^2$ iv d1, FU 400 mg/m$^2$ iv d1, FU 2,400 mg/m$^2$ continuous infusion 46 hours, LV 400 mg/m$^2$ iv d1, q2w.<br>Arm B: FU 400 mg/m$^2$ iv d1, FU 2,400 mg/m$^2$ continuous infusion 46 hours, LV 400 mg/m$^2$ iv d1, q2w. | PFS |
| Wang et al. 2015 [9] | FU/LV+Nal-IRI<br>FU/LV | Arm A: Nal-IRI 80 mg/m$^2$ iv d1, FU 2,400 mg/m$^2$ continuous infusion 46 hours, LV 400 mg/m$^2$ iv d1, q2w.<br>Arm B: FU 2,000 mg/m$^2$ continuous infusion 24 hours, d1 d8 d15 d22, LV 200 mg/m$^2$ iv d1 d8 d15 d22, q6w. | OS |
| Oettle et al. 2014 [10] | FU/LV+OXA<br>FU/LV | Arm A: OXA 85 mg/m$^2$, iv d8 d22, FU 2,000 mg/m$^2$ continuous infusion 24 hours, d1 d8 d15 d22, LV 200 mg/m$^2$ iv d1 d8 d15 d22, q6w.<br>Arm B: FU 2,000 mg/m$^2$ continuous infusion 24 hours, d1 d8 d15 d22, LV 200 mg/m$^2$ iv d1 d8 d15 d22, q6w. | OS |
| Se-Il et al. 2021 [11] | FU/LV+OXA+IRI<br>S-1 | Arm A: OXA 65 mg/m$^2$, iv d1, IRI 135 mg/m$^2$ iv d1, FU 1,000 mg/m$^2$ continuous infusion 24 hours, d1 d2, LV 400 mg/m$^2$ iv d1, q2w.<br>Arm B: S-1 80/100/120 mg/d based on BSA, po d1–28, q6w. | OS |

S-1: An oral agent of fluoropyrimidine. BSA: Body surface area. FU: Fluorouracil. LV: Leucovorin. OXA: Oxaliplatin. IRI: Irinotecan. Nal-IRI: Nanoliposomal irinotecan. PFS: Progression free survival. OS: Overall survival.

between trials. However, 2 of the trials did not define PFS and OS [9,11]. The frequencies of imaging scans to review tumor responses were not uniform between trials. 3 of the trials performed imaging scans every 6 weeks [8,9,11]. 2 of the trials performed imaging scans every 4 weeks [6,7]. 1 of the trials performed imaging scans every 2 months [10]. Only 2 of the trials illustrated that imaging scans to assess tumor responses were reviewed by independent review committee [6,7].

**Table 4. Methodological details which might cause bias of included studies.**

| References | Phase | Random | Blind | Randomization description | Withdraw description | ITT analysis | Multi-center | Jadad score |
|---|---|---|---|---|---|---|---|---|
| Ohkawa et al. 2015 [6] | II | Yes | No | Yes | Yes | Yes | Yes | 3 |
| Ioka et al. 2017 [7] | II | Yes | No | NC | Yes | Yes | Yes | 2 |
| Gill et al. 2016 [8] | III | Yes | No | NC | Yes | Yes | Yes | 2 |
| Wang et al. 2015 [9] | III | Yes | No | Yes | Yes | Yes | Yes | 3 |
| Oettle et al. 2014 [10] | III | Yes | No | Yes | Yes | Yes | Yes | 3 |
| Se-Il et al. 2021 [11] | III | Yes | No | Yes | Yes | Yes | Yes | 3 |

NC: No clear. ITT: Intend-to-treat.

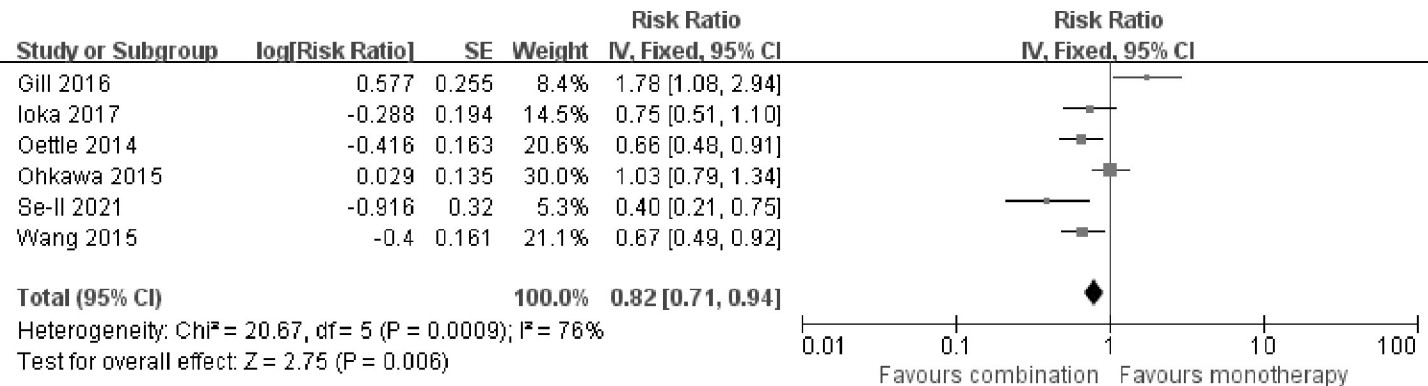

**Fig 2. Comparison of OS between fluoropyrimidine combination therapy and fluoropyrimidine monotherapy.** SE: Standard error. CI: Confidence interval. IV: Inverse variance. Diamonds represented the 95% confidence interval of overall effect.

## Overall survival

The impact of fluoropyrimidine combination therapy on OS was extracted directly from published data of the 6 included trials. The analysis showed that, fluoropyrimidine combination therapy improved OS compared with fluoropyrimidine monotherapy in patients with gemcitabine-refractory pancreatic cancer [HR 0.82 (0.71–0.94), p = 0.006]. But significant heterogeneity was found among the studies ($I^2$ = 76%, p = 0.0009) (**Fig 2**). When random-effect model was used, the final result of OS changed, which showed no statistical significance [HR 0.80 (0.59–1.09), p = 0.16]. Then we performed sensitivity analysis by excluding each study individually. The heterogeneity fluctuated between 62% and 80%.

In the subgroup analysis, fluoropyrimidine combined with irinotecan improved OS compared with fluoropyrimidine monotherapy [HR 0.70 (0.55–0.89), p = 0.004], without significant heterogeneity among studies ($I^2$ = 0%, p = 0.66). The HR for OS failed to show advantage in combining fluoropyrimidine with oxaliplatin [HR 1.03 (0.64–1.67), p = 0.90].

## Progression free survival

The impact of fluoropyrimidine combination therapy on PFS was extracted directly from the 6 included trials. The analysis showed that, fluoropyrimidine combination therapy improved PFS compared with fluoropyrimidine monotherapy [HR 0.71 (0.62–0.82), p<0.00001], with moderate heterogeneity among studies ($I^2$ = 55%, p = 0.05) (**Fig 3**). When random-effect model was used, the final result of PFS did not change [HR 0.70 (0.57–0.87), p = 0.001].

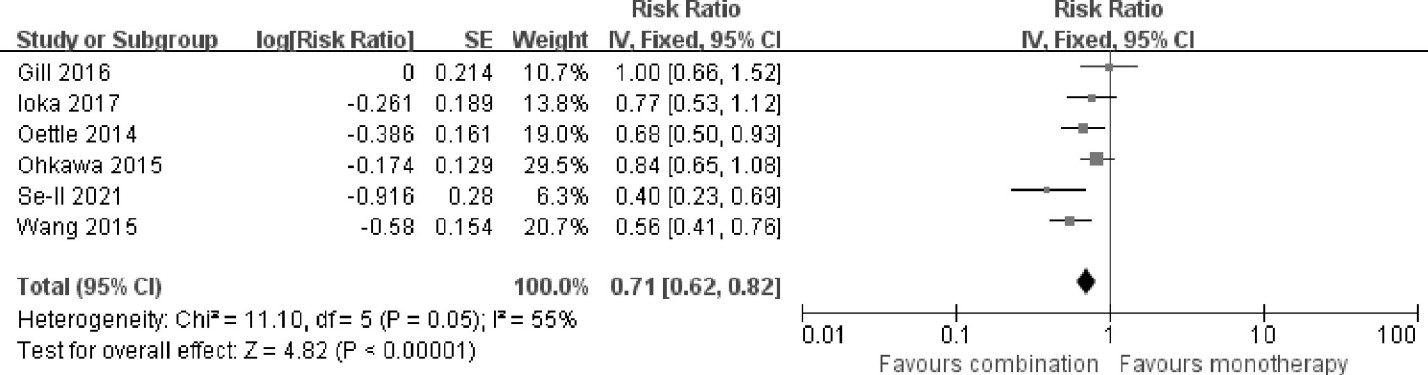

**Fig 3. Comparison of PFS between fluoropyrimidine combination therapy and fluoropyrimidine monotherapy.** SE: Standard error. CI: Confidence interval. IV: Inverse variance. Diamonds represented the 95% confidence interval of overall effect.

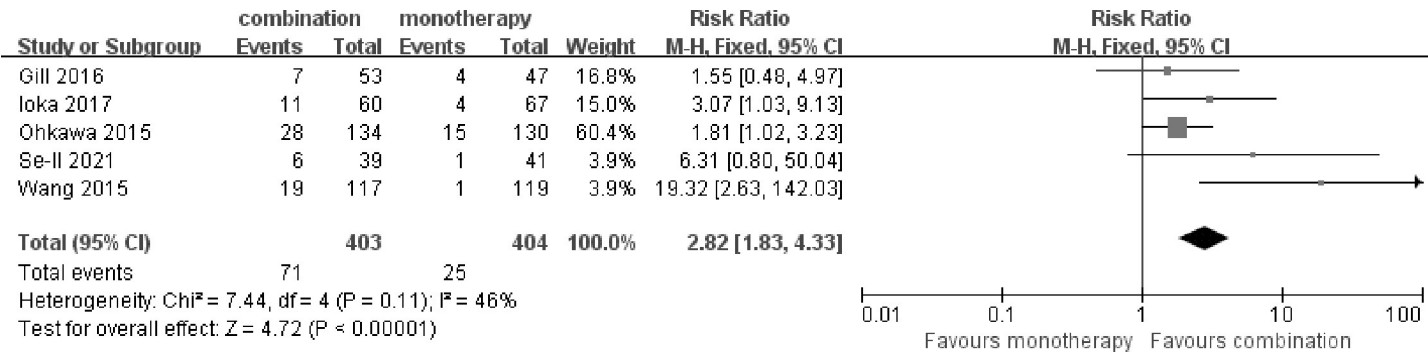

**Fig 4. Comparison of ORR between fluoropyrimidine combination therapy and fluoropyrimidine monotherapy.** CI: Confidence interval. M-H: Mantel-Haenszel. Diamonds represented the 95% confidence interval of overall effect.

In the subgroup analysis, fluoropyrimidine combined with irinotecan improved PFS [HR 0.64 (0.50–0.80), p = 0.0001], without significant heterogeneity ($I^2$ = 42%, p = 0.19). The HR for PFS also showed advantage in combining fluoropyrimidine with oxaliplatin [HR 0.81 (0.68–0.97), p = 0.02], without significant heterogeneity ($I^2$ = 11%, p = 0.33).

## Overall response rate

5 of the included trials described the impact of fluoropyrimidine combination therapy on ORR. The analysis revealed a significant difference between two arms. Fluoropyrimidine combination therapy improved ORR compared with fluoropyrimidine monotherapy [RR 2.82 (1.83–4.33), p<0.00001], without significant heterogeneity between studies ($I^2$ = 46%, p = 0.11) (**Fig 4**).

The subgroup analysis showed that, comparative analysis of RR for ORR showed advantage for the fluoropyrimidine combination therapy with oxaliplatin [RR 1.75 (1.05–2.95), p = 0.03]. The analysis for the regimen of fluoropyrimidine combined with irinotecan also showed advantage [RR 6.45 (2.54–16.39), p<0.0001].

## Toxicities

The outcome of the toxicities with grade≥3 for fluoropyrimidine combination therapy was assessed. Only certain toxicities were consistently described in the 6 articles. We assessed the toxicities of peripheral neuropathy mainly caused by oxaliplatin, toxicity of diarrhea mainly caused by irinotecan, and other common toxicities occurred in the routine chemotherapy procedure, for example, the nausea, vomiting, neutropenia and anemia. The analysis showed that the grade≥3 toxicities increased by the combination therapy were peripheral neuropathy [RR 5.18 (1.15–23.28), p = 0.03] ($I^2$ = 0%, p = 0.93), nausea [RR 2.31 (1.15–4.64), p = 0.02] ($I^2$ = 0%, p = 1.00), vomiting [RR 3.64 (1.54–8.61), p = 0.003] ($I^2$ = 0%, p = 0.97), and neutropenia [RR 3.74 (2.40–5.84), p<0.00001] ($I^2$ = 85%, p<0.0001) (**Fig 5**). In the subgroup analysis, grade≥3 diarrhea was observed in the regimens containing irinotecan [RR 2.41 (1.07–5.42), p = 0.03] ($I^2$ = 0%, p = 0.37).

## Publication bias

We used the highly sensitive search strategy to identify the relevant trials to minimize the potential of publication bias. And papers were collected strictly according to the inclusion criteria. Publication bias was detected by funnel plot. The statistical evidence of funnel plot symmetry was provided by Egger's test. No apparent publication bias was found (**Fig 6**). Egger's

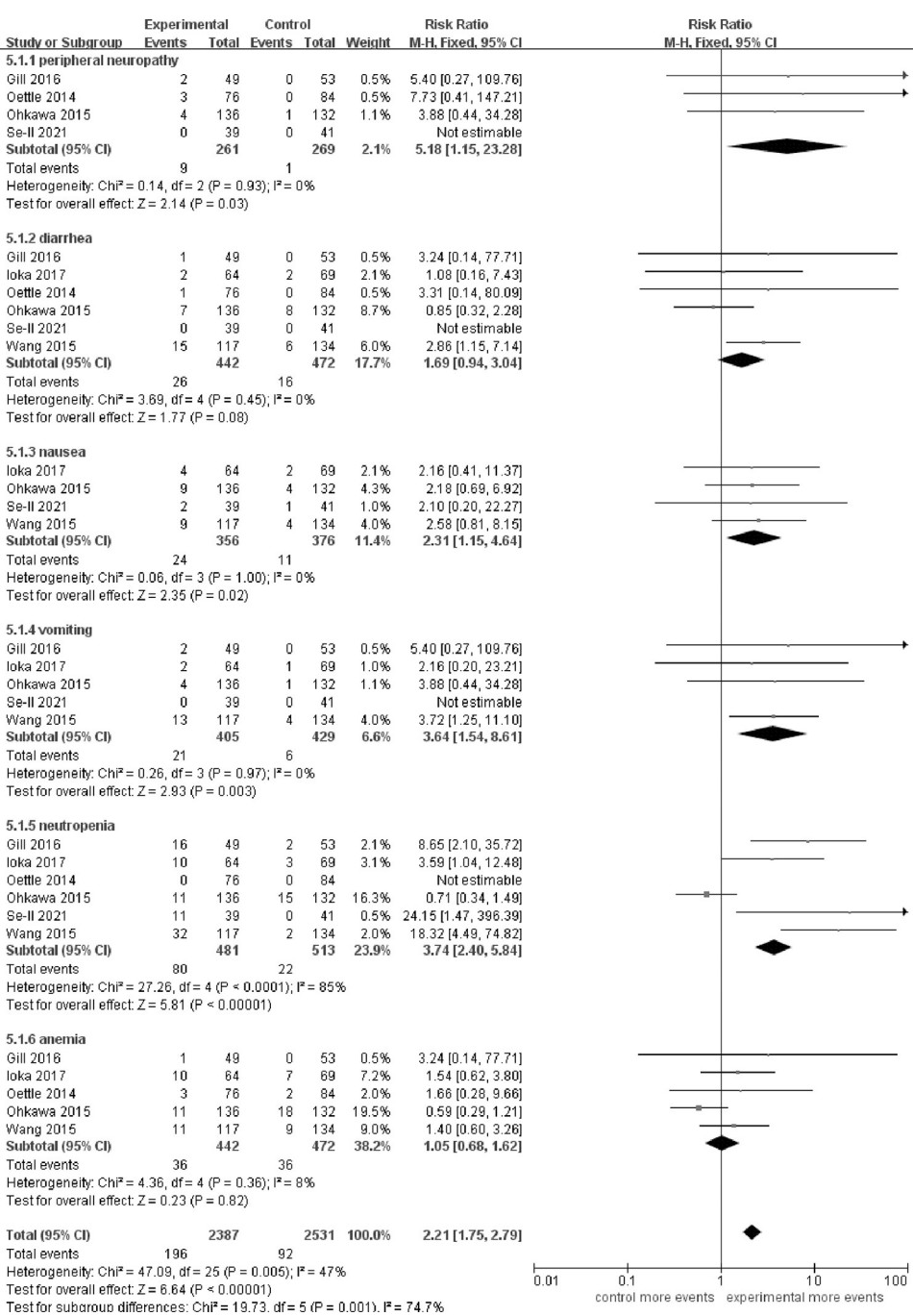

**Fig 5. Comparison of toxicities between fluoropyrimidine combination therapy and fluoropyrimidine monotherapy.** CI: Confidence interval. Diamonds represented the 95% confidence interval of overall effect.

test did not suggest any evidence of publication bias for OS (p = 0.79) and PFS (p = 0.49). However, the small number of the included studies limited the accuracy of the analysis.

## Discussion

Patients with advanced pancreatic cancer always have poor prognoses as a result of the high lethality of the disease. Compared with gemcitabine alone, combination therapy has obtained

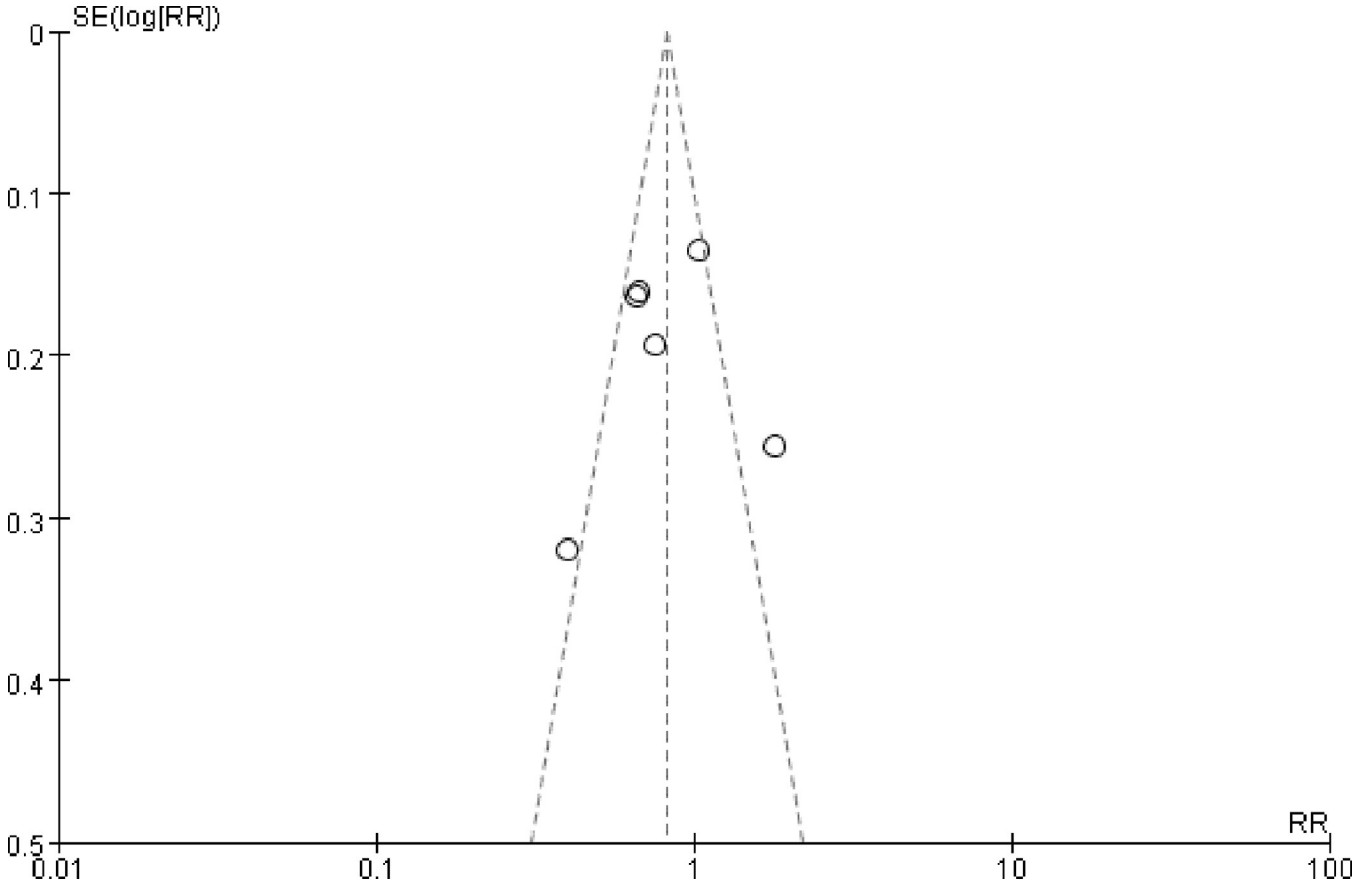

**Fig 6. Funnel plot to assess for evidence of publication bias.** Oblique line represented the pseudo 95% confidence interval. Circles represented the included studies.

significant survival benefits, which is mainly due to the introduction of new cytotoxic and targeted drugs in the past 20 years [12–14]. Patients with advanced pancreatic cancer are usually treated with first-line gemcitabine-based chemotherapy. There remains the problem of second-line setting in patients refractory to first-line gemcitabine-based treatment. Here, we conducted this systematic review and meta-analysis to evaluate the efficacy and safety profile of fluoropyrimidine combination therapy versus fluoropyrimidine monotherapy.

Our meta-analysis showed that fluoropyrimidine combination therapy increased ORR and PFS, in patients with gemcitabine-refractory pancreatic cancer. Fluoropyrimidine combination groups could prolong OS, but significant heterogeneity ($I^2$ = 76%) in the analysis of OS was found. This phenomenon might have been caused by the different administration regimens and baseline characteristics. The combination arm in Oettle 2014 used the OFF regimen, in which oxaliplatin was administered at a lower dose intensity on days 8 and 22 every 42 days [10]. Patients allocated to the combination arm in Ohkawa 2015 received oxaliplatin on day 1 every 21 days [6]. The dose intensity of oxaliplatin in these two regimens was lower than the administration of every 14 days in Gill 2016 [8]. In Gill 2016, the tolerability of the fluoropyrimidine monotherapy arm was remarkably better than the combination arm. The incidence of grade 3/4 adverse events in the combination arm was six-fold higher than that in the monotherapy arm (63% v 11%). Thus, the reduction of oxaliplatin intensity might lead to better

tolerance in the combination regimen. The randomized phase 2 trial conducted by Yoo et al compared modified version of FOLFOX (folinic acid, fluorouracil, and oxaliplatin) regimen for treatment of gemcitabine-refractory advanced pancreatic cancer, in which oxaliplatin was administered on day 1 every 14 days [15]. However, the median overall survival was only 3.5 months in this study. All these indicated the considerable importance of the second-line treatment tolerance in patients with advanced pancreatic cancer, which might influence the final results of overall survival. Due to concerns regarding toxicity in second-line treatment, it should be carefully considered to use administration of oxaliplatin every 14 days in frail patients with pancreatic cancer.

The analyses of adverse events were slightly different between groups, indicating that both therapeutic regimens were well tolerated. Incidence of grade≥3 peripheral sensory neuropathy, the most critical toxicity of oxaliplatin, was low (4.1%, 3.9% and 0.8%) in the 3 trials included in this analysis. In the treatment of metastatic colorectal cancer, when the cumulative dose of oxaliplatin ranging from 765 mg/m$^2$ to 1020 mg/m$^2$, 30% of patients will experience grade≥3 neuropathy induced by oxaliplatin [16–18]. And the less cumulative dose of oxaliplatin led to less frequency of grade≥3 neuropathy in the second line setting in pancreatic cancer. In our analysis, grade≥3 diarrhea was observed in the regimen of fluoropyrimidine combined with irinotecan. In Wang 2015, the severe gastrointestinal events caused by nanoliposomal irinotecan monotherapy at a higher dose (120 mg/m$^2$) every 3 weeks were much more common than that caused by nanoliposomal irinotecan combination therapy at a lower dose (80 mg/m$^2$) every 2 weeks. The mFOLFIRINOX regimen in Se-Il 2021 [11] comprised intravenous oxaliplatin (65 mg/m$^2$), irinotecan (135 mg/m$^2$), leucovorin (400 mg/m$^2$), and continuous infusion of 5-FU (2000 mg/m$^2$ for 2 days). We found the drug dosages in this study were more attenuated than those used in routine practice. But the frequencies of grade≥3 neutropenia were still high (28%). Furthermore, one-third of patients treated with the regimen of mFOLFIRINOX required dose reduction during treatment. But the attenuated dosages of drugs and timely adjustment might have prevented discontinuation of treatment, and brought clinical benefits to patients with advanced pancreatic cancer.

There were several limitations that should be taken into account in this analysis. First of all, although no apparent publication bias was found by funnel plots and Egger's test, the small number of the included studies limited the power of the analysis. Second, heterogeneity was found among the included studies due to different treatment regimens and baseline characteristics. Third, the frequency of imaging examination between different studies was not consistent, which might affect the final results of ORR. Fourth, we found that the drug dosages might have important impact on the OS of patients, so the subgroup analysis such as age would be needed to make the conclusion stronger. 5 of the included studies performed subgroup analysis by age [6–9,11], but the dividing points of age were not uniform. Some studies used the age of 65 as the dividing point [6,7,9,11], while Gill 2016 used the age of 70 [8]. Finally, all the 6 trials included in this analysis were open label, which might have caused certain bias. These limitations indicated the urgent need to design clinical trials using standardized, unbiased methods and a larger sample size to confirm the efficacy and safety of fluoropyrimidine combination regimens.

Our study focused on the fluoropyrimidine-based therapy in second-line treatment of gemcitabine-refractory pancreatic cancer. We found that the dosage of chemotherapy drug added to fluoropyrimidine might be one of the important factors affecting the prognosis of patients. In conclusion, fluoropyrimidine combination therapy showed advantage compared with fluoropyrimidine monotherapy. The identification of reasonable dose intensity of chemotherapy drugs is warranted in future trials to bring better clinical benefits.

## Supporting information

**S1 Table. PRISMA checklist.**
(DOCX)

**S1 Appendix. PRISMA flow diagram.**
(DOC)

## Acknowledgments

We thank Daidi Fu for her helpful advice in paper writing.

## Author Contributions

**Conceptualization:** Wei Tian, Rui Wang.

**Data curation:** Wei Tian, Lina Zhang, Xiao Liu, Rui Wang.

**Formal analysis:** Wei Tian, Xiao Liu, Rui Wang.

**Methodology:** Xiao Ma.

**Resources:** Xiao Ma.

**Software:** Lina Zhang.

**Supervision:** Xiao Liu.

**Validation:** Lina Zhang.

**Writing – original draft:** Wei Tian, Rui Wang.

**Writing – review & editing:** Xiao Ma, Rui Wang.

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
