## [Decision Letter · Decision Letter 0]

5 Dec 2022

PONE-D-22-27493Fluoropyrimidine combination therapy versus fluoropyrimidine monotherapy for gemcitabine-refractory advanced pancreatic cancer: a meta-analysis of randomized controlled trialsPLOS ONE

Dear Dr. Wang,

Thank you for submitting your manuscript to PLOS ONE. After careful consideration, we feel that it has merit but does not fully meet PLOS ONE’s publication criteria as it currently stands. Therefore, we invite you to submit a revised version of the manuscript that addresses the points raised during the review process.

We look forward to receiving your revised manuscript.

Kind regards,

Alberto Meyer, MD, PhD

Academic Editor

PLOS ONE

Journal Requirements:

2. Please identify your study as "systematic review and meta-analysis" in the title.

3. Thank you for submitting the above manuscript to PLOS ONE. During our internal evaluation of the manuscript, we found significant text overlap between your submission and previous work in the Abstract, Methods and Discussion section. We would like to make you aware that copying extracts from previous publications, especially outside the methods section, word-for-word is unacceptable. In addition, the reproduction of text from published reports has implications for the copyright that may apply to the publications. Please revise the manuscript to rephrase the duplicated text, cite your sources, and provide details as to how the current manuscript advances on previous work. Please note that further consideration is dependent on the submission of a manuscript that addresses these concerns about the overlap in text with published work. We will carefully review your manuscript upon resubmission and further consideration of the manuscript is dependent on the text overlap being addressed in full. Please ensure that your revision is thorough as failure to address the concerns to our satisfaction may result in your submission not being considered further.

Additional Editor Comments:

Wang et al report a study-level meta-analysis of 6 RCTs, in second-line treatment of pancreas cancer, comparing fluoropyrimidine alone (either 5FU or S-1; note capecitabine was not included), with the same FP in addition to another chemotherapy agent (oxaliplatin (2 studies), irinotecan (2 studies), Nal-Iri – a modified formulation of irinotecan (one study) – or oxaliplatin plus irinotecan (one study)).

The MA design is reasonable and the search seems to have been done well, revealing only 6 eligible studies.

There is limited information in this paper on the characteristics of each of the 6 papers, and the patients therein. For example, there is not a description of how each study defined “gemcitabine refractory advanced pancreatic cancer” , whether some were limited to disease progression during gemcitabine treatment, whether a time limit after gem treatment was specified as part of a definition of gem-refractory disease.

Revise.

REgards,

Reviewers' comments:

Reviewer's Responses to Questions

**Comments to the Author**

1. Is the manuscript technically sound, and do the data support the conclusions?

Reviewer #1: Yes

Reviewer #2: Partly

Reviewer #3: Partly

2. Has the statistical analysis been performed appropriately and rigorously? 

Reviewer #1: Yes

Reviewer #2: I Don't Know

Reviewer #3: I Don't Know

3. Have the authors made all data underlying the findings in their manuscript fully available?

Reviewer #1: Yes

Reviewer #2: Yes

Reviewer #3: Yes

4. Is the manuscript presented in an intelligible fashion and written in standard English?

Reviewer #1: Yes

Reviewer #2: Yes

Reviewer #3: Yes

5. Review Comments to the Author

Reviewer #1: 1、The last date of the article screen is January 2022, please update the most recent research.

2、Please specify in the article which guides Jadad score is based on.

3、The results of Egger’s test should be added in detail in the “Publication bias”.

4、The content of the article needs to be examined carefully to avoid some grammatical mistakes.

Reviewer #2: In this study “Fluoropyrimidine combination therapy versus fluoropyrimidine monotherapy for gemcitabine”, the authors conducted a meta-analysis study of RCTs to compare fluropyrimidine based therapy. While studies have been accumulating with such evidence, similar review of meta-analysis have been reported by Yang et al 2015 that included demographic data, treatment response, objective response rate (ORR), progression-free and overall survival (PFS and OS, respectively), and toxicities.

While it is interesting, this meta-analysis provides just another addition to the study pool but derived from just a handful of 6 study trials. As such comparative meta-analysis study from larger study trials with multiple locations, demographic group, age etc will make the conclusion stronger. Although, the written text is alright I find the result difficult to compare with the poor-quality figures presented with no legends and explanation.

I encourage the authors to include all abbreviations, legends, labels and explanation of the figures and here’s my comments-

Table1: Please explain the abbreviation used for search strategy. Readers find it difficult in understating what is meant by “$” or “mp” etc. While this may be familiar for clinical researchers, simply putting a table of keyword used for the search lacks clarity.

Please provide high resolution figure and all the figure legends

Fig 6- Label axes properly and explain the funnel plot in legend, what the circles meant etc

Table2- What is S-1?

Reviewer #3: PONE -D- 22-27493 – Wang et al - MA of FP alone vs FP-combination

Wang et al report a study-level meta-analysis of 6 RCTs, in second-line treatment of pancreas cancer, comparing fluoropyrimidine alone (either 5FU or S-1; note capecitabine was not included), with the same FP in addition to another chemotherapy agent (oxaliplatin (2 studies), irinotecan (2 studies), Nal-Iri – a modified formulation of irinotecan (one study) – or oxaliplatin plus irinotecan (one study)).

Overall Comments:

The subject of this MA is relevant, as recurrent pancreas cancer after first line chemotherapy is a serious issue, now with several possible treatments. The MA design is reasonable and the search seems to have been done well, revealing only 6 eligible studies.

The first author does not seem to have provided an ORCID.

There are multiple minor grammatical errors throughout, such as Line 126: are shown, and Line 143 uses the incorrect tense.

More information about how data was analysed would be valuable. For example, were medians compared? How were confidence intervals managed or data ranges managed?

There are already 4 papers titled meta-analysis in the 2L treatment of pancreatic cancer (Lu W, J Int Med Res, 2022; Wainberg Z BMC Cancer 2020; Sonbol MB, Cancer 2017; Zhong S, Medicine (Baltimore) 2017.) What will this paper add to the literature?

Major Comments:

There is limited information in this paper on the characteristics of each of the 6 papers, and the patients therein. For example, there is not a description of how each study defined “gemcitabine refractory advanced pancreatic cancer” , whether some were limited to disease progression during gemcitabine treatment, whether a time limit after gem treatment was specified as part of a definition of gem-refractory disease.

The description of patient characteristics is not as extensive as, for example, in the tables of the Ohkawa paper.

The primary and secondary outcomes, and how each study defined them, are not explained in detail. How were PFS and ORR defined? (were there differences in definition between papers?). ORR requires an assessment of response, usually with CT scans, but the frequency and veracity of scans (independent expert reporting, or a potentially biased investigator) can affect the ORR.

Since this is a retrospective study-level meta-analysis, it is possible that data elements are recorded differently in each paper, and that some data elements are missing. There is no consideration in Methods about how missing or “difficult to determine” data was dealt with. How did the 2 data collectors work together? How were inconsistencies, missing data and differences in interpretation dealt with?

Is there missing data in any of the studies (there is some data inconsistencies in the paper of Ohkawa)?

What is the duration of followup (median and range) in each paper and is it important?

There is not data about dose-intensity for each study – the doses referred to are intended; there is no consideration of doses actually received, or to duration of treatment in each study. This may be important as oxaliplatin neuropathy is a function of total dose over time.

I note Wang-Gillam et al has 117 and 149 patients in the 2 arms of their study; this suggests that there was some imbalance in the randomisation. This is not discussed.

Two studies are called phase II and 4 are called phase III – What is the difference here?

Table 3 has a column called “allocation concealment” (4 yes, 2 not clear) – what does this mean? I note that all studies are said in the text to be “open label” – this usually means there is no blinding of treatment so no allocation concealment.

Minor Comments:

Abstract:

the second sentence in Conclusion is not based on any data in the rest of the abstract. The abstract needs revision after the paper itself is revised, to ensure consistency and so the Abstract can stand alone but still be a fair summary of the paper.

Figures:

The Forest plots need more annotation and the terms which appear need to be all defined, including IV and M-H. What does the size and length of the diamonds represent. Why is the number at risk or observed/expected not reports in the figures?

The direction of the figure (favours monotherapy vs favours combination) is switched between fig 3 and 4. These should be consistent.

Fig 4 has a typographical error (momotherapy).

6. PLOS authors have the option to publish the peer review history of their article (what does this mean?). If published, this will include your full peer review and any attached files.

Reviewer #1: No

Reviewer #2: No

Reviewer #3: No

---

## [Author Response · Author response to Decision Letter 0]

16 Jan 2023

January16, 2023

Reply for manuscript “Fluoropyrimidine combination therapy versus fluoropyrimidine monotherapy for gemcitabine-refractory advanced pancreatic cancer: a meta-analysis of randomized controlled trials”

 Reply to editor:

Dear editor:

Thank you for providing us with opportunity to submit the revised manuscript. We would like to express our sincere gratitude to all the reviewers for their constructive comments. According to the helpful comments, we have extensively revised the manuscript by correcting mistakes and supplementing the required materials to make our manuscript better. At the same time, we ensure that the manuscript meets PLOS ONE's style requirements.

Journal Requirements:

Answer: We carefully checked the manuscript to make it meet the requirements.

2. Please identify your study as "systematic review and meta-analysis" in the title.

Answer: We added the “systematic review” in the title and abstract in the manuscript. Line 2, 4, 38.

3. Thank you for submitting the above manuscript to PLOS ONE. During our internal evaluation of the manuscript, we found significant text overlap between your submission and previous work in the Abstract, Methods and Discussion section. We would like to make you aware that copying extracts from previous publications, especially outside the methods section, word-for-word is unacceptable. In addition, the reproduction of text from published reports has implications for the copyright that may apply to the publications. Please revise the manuscript to rephrase the duplicated text, cite your sources, and provide details as to how the current manuscript advances on previous work. Please note that further consideration is dependent on the submission of a manuscript that addresses these concerns about the overlap in text with published work. We will carefully review your manuscript upon resubmission and further consideration of the manuscript is dependent on the text overlap being addressed in full. Please ensure that your revision is thorough as failure to address the concerns to our satisfaction may result in your submission not being considered further.

Answer: We revised the parts of text overlap in the manuscript. 

Answer: We revised the Data Availability Statement in the cover letter. All relevant data are within the paper and its Supporting Information files.

Answer: We added the captions of Supporting Information Files at the end of the manuscript. Line 396-398

Additional Editor Comments:

Wang et al report a study-level meta-analysis of 6 RCTs, in second-line treatment of pancreas cancer, comparing fluoropyrimidine alone (either 5FU or S-1; note capecitabine was not included), with the same FP in addition to another chemotherapy agent (oxaliplatin (2 studies), irinotecan (2 studies), Nal-Iri – a modified formulation of irinotecan (one study) – or oxaliplatin plus irinotecan (one study)).

Answer: There was no eligible study about capecitabine.

The MA design is reasonable and the search seems to have been done well, revealing only 6 eligible studies.

There is limited information in this paper on the characteristics of each of the 6 papers, and the patients therein. For example, there is not a description of how each study defined “gemcitabine refractory advanced pancreatic cancer” , whether some were limited to disease progression during gemcitabine treatment, whether a time limit after gem treatment was specified as part of a definition of gem-refractory disease. 

Answer: We added the median age of patients and the descriptions of “gemcitabine refractory” in Table 2.

Reply to reviewers:

Dear reviewer:

We are really grateful to you for your time spending on our manuscript and making helpful comments. Those comments are all valuable and helpful to make our manuscript better. We discussed each of the comments individually along with our responses and marked the line number in the Revised Manuscript with Track Changes. We hope you will find the revised version more satisfactory. We are more than happy to make any further revision that will improve the manuscript. 

Reply to reviewer 1:

Reviewer #1: 1、The last date of the article screen is January 2022, please update the most recent research. 

Answer: We searched the databases and found no latest eligible research. 

2、Please specify in the article which guides Jadad score is based on. 

Answer: We added the definition of Jadad score used in our analysis in the manuscript. Line 134-140.

3、The results of Egger’s test should be added in detail in the “Publication bias”. 

Answer: We performed Egger’s test using stata and added the results in the manuscript. Line 49-50, Line 249-251.

4、The content of the article needs to be examined carefully to avoid some grammatical mistakes. 

Answer: We examined the manuscript carefully and revised the mistakes.

Reviewer #2: In this study “Fluoropyrimidine combination therapy versus fluoropyrimidine monotherapy for gemcitabine”, the authors conducted a meta-analysis study of RCTs to compare fluropyrimidine based therapy. While studies have been accumulating with such evidence, similar review of meta-analysis have been reported by Yang et al 2015 that included demographic data, treatment response, objective response rate (ORR), progression-free and overall survival (PFS and OS, respectively), and toxicities.

While it is interesting, this meta-analysis provides just another addition to the study pool but derived from just a handful of 6 study trials. As such comparative meta-analysis study from larger study trials with multiple locations, demographic group, age etc will make the conclusion stronger. Although, the written text is alright I find the result difficult to compare with the poor-quality figures presented with no legends and explanation. 

I encourage the authors to include all abbreviations, legends, labels and explanation of the figures and here’s my comments.

Reply to Reviewer 2:

Table1: Please explain the abbreviation used for search strategy. Readers find it difficult in understating what is meant by “$” or “mp” etc. While this may be familiar for clinical researchers, simply putting a table of keyword used for the search lacks clarity. 

Answer: We added the explanation of the abbreviations used in the search strategy in the legend of Table 1.

Please provide high resolution figure and all the figure legends. 

Answer: We added all the abbreviations in the figure legends. 

Fig 6- Label axes properly and explain the funnel plot in legend, what the circles meant etc 

Answer: We added the explanations of oblique line and circles in the legend of Fig 6. 

Table2- What is S-1? 

Answer: We added the explanation of S-1 in the legend of Table 2.

Reviewer #3: PONE -D- 22-27493 – Wang et al - MA of FP alone vs FP-combination

Wang et al report a study-level meta-analysis of 6 RCTs, in second-line treatment of pancreas cancer, comparing fluoropyrimidine alone (either 5FU or S-1; note capecitabine was not included), with the same FP in addition to another chemotherapy agent (oxaliplatin (2 studies), irinotecan (2 studies), Nal-Iri – a modified formulation of irinotecan (one study) – or oxaliplatin plus irinotecan (one study)).

Overall Comments:

The subject of this MA is relevant, as recurrent pancreas cancer after first line chemotherapy is a serious issue, now with several possible treatments. The MA design is reasonable and the search seems to have been done well, revealing only 6 eligible studies.

Reply to Reviewer 3:

The first author does not seem to have provided an ORCID. 

Answer: The ORCID of the first author was “https://orcid.org/0000-0002-8214-5347”.

There are multiple minor grammatical errors throughout, such as Line 126: are shown, and Line 143 uses the incorrect tense. 

Answer: We examined the manuscript carefully and revised the mistakes.

More information about how data was analysed would be valuable. For example, were medians compared? How were confidence intervals managed or data ranges managed?

Answer: The hazard ratio (HR) of time-to-event data for OS and PFS, and the risk ratio (RR) of dichotomous data for ORR and toxicities with 95% confidence interval were extracted directly from published data of the 6 included trials. 

There are already 4 papers titled meta-analysis in the 2L treatment of pancreatic cancer (Lu W, J Int Med Res, 2022; Wainberg Z BMC Cancer 2020; Sonbol MB, Cancer 2017; Zhong S, Medicine (Baltimore) 2017.) What will this paper add to the literature? 

Answer: Our study focused on the fluoropyrimidine-based therapy in second-line treatment of gemcitabine-refractory pancreatic cancer. We found that the dosage of chemotherapy drug added to fluoropyrimidine might be one of the important factors affecting the prognosis of patients. We added this part in the discussion. Line 329-331.

Major Comments:

There is limited information in this paper on the characteristics of each of the 6 papers, and the patients therein. For example, there is not a description of how each study defined “gemcitabine refractory advanced pancreatic cancer” , whether some were limited to disease progression during gemcitabine treatment, whether a time limit after gem treatment was specified as part of a definition of gem-refractory disease. 

Answer: We added the descriptions of “gemcitabine refractory” in Table 2. This is part of the characteristics of patients.

The description of patient characteristics is not as extensive as, for example, in the tables of the Ohkawa paper. 

Answer: We added the median age of patients and the descriptions of “gemcitabine refractory” in Table 2.

The primary and secondary outcomes, and how each study defined them, are not explained in detail. How were PFS and ORR defined? (were there differences in definition between papers?). ORR requires an assessment of response, usually with CT scans, but the frequency and veracity of scans (independent expert reporting, or a potentially biased investigator) can affect the ORR. 

Answer: We added the definitions of OS and PFS in the part of “Characteristics of included studies”. The frequencies of imaging scans in the included studies were also added. The frequency of imaging examination between different studies was not consistent. And we discussed it in the “limitation part” of the manuscript. Line 171-178. Line 318-319.

Since this is a retrospective study-level meta-analysis, it is possible that data elements are recorded differently in each paper, and that some data elements are missing. There is no consideration in Methods about how missing or “difficult to determine” data was dealt with. How did the 2 data collectors work together? How were inconsistencies, missing data and differences in interpretation dealt with? 

Answer: We added the descriptions of how to obtain missing data and how to deal with the discrepancies. Line 123-125. Line 139-140.

Is there missing data in any of the studies (there is some data inconsistencies in the paper of Ohkawa)? 

Answer: We checked the data in the studies and found no missing data.

What is the duration of followup (median and range) in each paper and is it important? 

Answer: All the included trials set the minimum number of events that were required for the analysis of detecting the assumed differences. As a result, duration of follow up was not mentioned in the trials.

There is not data about dose-intensity for each study – the doses referred to are intended; there is no consideration of doses actually received, or to duration of treatment in each study. This may be important as oxaliplatin neuropathy is a function of total dose over time. 

Answer: We added the dosages and dose-intensities of drugs for each study in Table 3.

I note Wang-Gillam et al has 117 and 149 patients in the 2 arms of their study; this suggests that there was some imbalance in the randomisation. This is not discussed. 

Answer: We added the reason why the number of patients was imbalance in the trial. Line 164-167.

Two studies are called phase II and 4 are called phase III – What is the difference here?

Answer: All the included studies did not mention the definitions of “phase”, so we did not know the exactly difference between them.

Table 3 has a column called “allocation concealment” (4 yes, 2 not clear) – what does this mean? I note that all studies are said in the text to be “open label” – this usually means there is no blinding of treatment so no allocation concealment. 

Answer: We revised “allocation concealment” to “Randomization description” in the table. Table 4.

Minor Comments:

Abstract:

the second sentence in Conclusion is not based on any data in the rest of the abstract. The abstract needs revision after the paper itself is revised, to ensure consistency and so the Abstract can stand alone but still be a fair summary of the paper. 

Answer: We revised the part of “conclusion” in the abstract. Line 58-64.

Figures:

The Forest plots need more annotation and the terms which appear need to be all defined, including IV and M-H. What does the size and length of the diamonds represent. Why is the number at risk or observed/expected not reports in the figures?

Answer: We added the annotations of “CI, IV, M-H and diamonds” in the legend of figures. The number at risk was reported in Kaplan-Meier curves in the clinical trials. But the forest plot in meta-analysis did not have this element.

The direction of the figure (favours monotherapy vs favours combination) is switched between fig 3 and 4. These should be consistent. 

Answer: The Risk Ratio for ORR was greater than 1.0，so it located in the right side of the forest plot. But the HR for OS and PFS was less than 1.0, so it located in the left side of the forest plot. This was the reason why the direction of the figure (favours monotherapy vs favours combination) was switched between fig 3 and 4.

Fig 4 has a typographical error (momotherapy). 

Answer: We revised it to “monotherapy”. Fig 4.

---

## [Editor Report · Decision Letter 1]

14 Feb 2023

Fluoropyrimidine combination therapy versus fluoropyrimidine monotherapy for gemcitabine-refractory advanced pancreatic cancer: a systematic review and meta-analysis of randomized controlled trials

PONE-D-22-27493R1

Dear Dr. Wang,

We’re pleased to inform you that your manuscript has been judged scientifically suitable for publication and will be formally accepted for publication once it meets all outstanding technical requirements.

Kind regards,

Alberto Meyer, MD, PhD

Academic Editor

PLOS ONE

Additional Editor Comments (optional):

Dear Authors,

Thank you for accepting our recommendations for revision and incorporating the relevant changes.

I am satisfied with your response and thus happy to recommend in favour of publication of your study.

Kind regards
---

## [Editor Report · Acceptance letter]

21 Feb 2023

PONE-D-22-27493R1 

Fluoropyrimidine combination therapy versus fluoropyrimidine monotherapy for gemcitabine-refractory advanced pancreatic cancer: a systematic review and meta-analysis of randomized controlled trials 

Dear Dr. Wang:

I'm pleased to inform you that your manuscript has been deemed suitable for publication in PLOS ONE. Congratulations! Your manuscript is now with our production department. 

Kind regards, 

on behalf of

Professor Alberto Meyer 

Academic Editor

PLOS ONE